# On the Optimal Whole-Body Vibration Protocol for Muscle Strength

Abdullah Al Masud [1] 📀, Chwan-Li Shen [2] 📀 and Ming-Chien Chyu [1,3,*]

1 Department of Mechanical Engineering, Texas Tech University, Lubbock, TX 79409, USA
2 Department of Pathology, School of Medicine, Texas Tech University Health Sciences Center, Lubbock, TX 79409, USA
3 Graduate Healthcare Engineering Program, Texas Tech University, Lubbock, TX 79409, USA
* Correspondence: m.chyu@ttu.edu

**Abstract:** The application of Whole Body Vibration (WBV) has been demonstrated to be effective in improving muscle strength/power by a number of studies, but an optimal training protocol has never been established. This paper presents a review of studies on the effects of WBV on muscles and an analysis of data to identify the optimal protocols for the most beneficial neuromuscular responses in terms of vibration frequency, amplitude, knee flexion angle, body posture (standing, sitting, supine, prone), muscle type (quadriceps, hamstrings), and vibration mode (superoinferior, anteroposterior, rotational). Ninety articles were selected for final review from initially selected 2093 articles using PRISMA guidelines. The findings suggest that the beneficial effects of WBV increase with frequency and amplitude but the optimal frequency and amplitude have not been established. The effect of the knee flexion angle is not clear. The optimal WBV protocol should be determined by considering the adverse effects of WBV on all parts of the human body including that related to head acceleration. WBV in sitting or lying positions may provide a better muscle response than standing. Directions for future research are discussed with regard to establishing the optimal WBV protocol as a safe and effective therapeutic/exercise modality for improving muscle strength and health.

**Keywords:** neuromuscular; muscle strength; optimal; therapeutic; vibration training; whole-body vibration



## 1. Introduction

Whole Body Vibration (WBV) has become a topic of extensive research for its potential effects on improving muscle strength [1] and power [2]. Evidence shows that vibratory stimulus can significantly improve neuromuscular activity, as confirmed by studies related to surface electromyography (sEMG) [3–5]. A possible mechanism is the stretch reflex response caused by the initiation of alpha-motoneurons and the stretch of muscle spindles, leading to tonic vibration reflex (TVR) [3–8]. Discrepancies have been reported due to differences in vibration type, frequency, amplitude, exposure time, and posture of subjects [9–11]. Some other factors such as load-carrying during the vibration training [12,13] and exercises performed before or during the vibration training [14,15] were found to improve muscle strength and physical performance. An increase in muscle activity and gravitational load due to vibration can also change hormone concentrations as well as neurogenic/morphological adaptation [4,8]. On the other hand, high-frequency WBV may result in adverse effects such as intervertebral disc displacement, hearing loss, and visual impairment [16]. However, vibrations with low frequency and low intensity are not effective in terms of the desired muscle health outcomes. Therefore, there is a need to investigate the optimal WBV training protocol for the most beneficial muscular health outcomes in terms of neuromuscular performances with the lowest risk of negative side effects that may not be related to the muscles. Most clinical studies only reported the results of certain WBV protocols implemented on human subjects without scientific justification why

such vibration frequency, amplitude, method of application, etc., are employed. Although some studies [17–21] suggested optimal vibration protocols based on clinical trials, true optimization cannot be achieved by discrete and fixed values of parameters assigned to different groups tested in clinical trials.

Through a narrative review, the present study aims to analyze the results of clinical trials in which WBV was applied to the feet of human subjects in both standing and sitting positions, as well as other postures (supine, prone), in an attempt to identify the optimal WBV intervention protocol that is safe while maximizing improved muscle strength. Clinical studies related to the effects of WBV on muscles in standing posture are discussed in Section 3.1 and with postures other than standing in Section 3.2. A review of the adverse effects of WBV is presented in Section 3.3, and a review of the literature concerning the optimal WBV protocol is presented in Section 3.4. An approach to establishing an optimal WBV protocol in terms of vibration frequency, amplitude, knee flexion angle, body posture, and muscle type, as well as future research, is discussed in Section 4, and the conclusion is presented in Section 5.

## 2. Methods

The present literature search was performed in two steps: (1) the use of electronic databases for published work; (2) a manual search of the reference lists of primary studies identified for this review. Selected publications were screened based on the inclusion and exclusion criteria to identify relevant studies.

### 2.1. Literature Search Strategy

The literature search of this review was based on PubMed, Scopus, Engineering Village, Science Direct, Cinahl, Cochrane, and Web of Science using different combinations of the following keywords: "Whole Body Vibration", "vibration therapy", "vibration training", "muscle strength", "muscle power", "muscle activity", "neuromuscular", "electromyography", "EMG", and "frequency". This review covers papers since 1998 as major research progress has been made in the field of WBV since that time. The present literature search included randomized control trials, pilot studies, and other types of papers that were (a) published in English, (b) related to WBV alone or a combination of WBV and any other exercise, (c) with the outcome(s) related to muscle power/strength, and (d) full-length articles. Articles related to local vibration therapy were excluded.

### 2.2. Inclusion and Exclusion Criteria

To be included in the present study, articles need to be related to WBV's effects on muscles in healthy humans. Excluded from this study are WBV studies investigating subjects with neuromuscular disorders (such as stroke, cerebral palsy, and sclerosis), studies on direct vibration applied to certain muscles, studies with inappropriate study design, with fewer than 5 subjects, or studies without defined vibration parameters. Articles in press, dissertations, and conference proceedings were also excluded from this review.

### 2.3. Results of Literature Search

As depicted in Figure 1, the flow chart of the literature review, 2093 articles were found from the initial literature search, and 90 articles were finally selected.

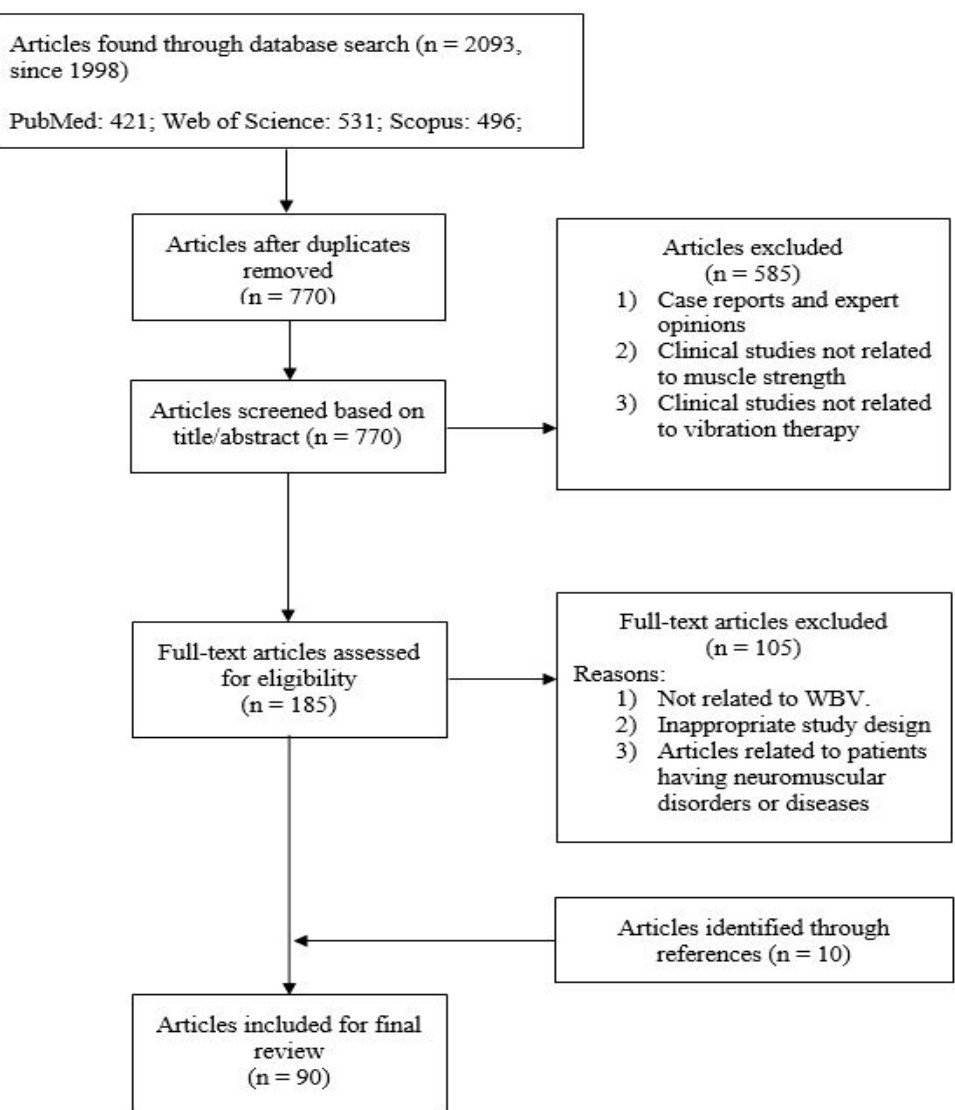

**Figure 1.** Flow chart for the literature review.

## 3. Results

### 3.1. Clinical Studies on Effects of Whole Body Vibration on Muscle in Standing Position

Published clinical studies and existing reviews [22–24] indicate that WBV can bring forth improvement in muscle strength, power, and flexibility. The most recent reviews by Bemben et al. [23] and Alam et al. [24] focused on effects of WBV on improved neuromuscular performance in elderly and young populations, considering the outcomes of the clinical trials. Effects of WBV on muscles have been investigated in a number of clinical studies where the whole body is subjected to vibration by a platform at the feet, with different levels of vibration frequency, amplitude, types of vibration and methods of application, intervention durations, numbers of repetition, and knee flexion angles during WBV. Most of the studies featured frequencies ranging from 15 to 60 Hz, amplitudes from 2 to 6 mm, and 2 to 4 sessions of training per week [25–34] with a variety of the outcome measurement parameters reported, as summarized in Table S1 for healthy young adults (18–32 years of age) and Table S2 for the elderly (54–82 years of age). Although published clinical studies used both elderly and young participants to investigate the effects of vibration training, differences in muscle strength between different age groups have not been well studied. Perchthaler et al. [31] found 18.55% increase in counter-movement jump height in the elderly, whereas there was only 3–4.5% increase in the same outcome measure in the young adults after vibration training [35–38].

Muscle strength and power reduce with age [37]. Reduced muscle power and functional ability increase the risk of fall and negatively affect daily activities in the elderly. Although resistance training can counteract age-related muscle strength loss [39], it is challenging for the aged population to participate in such training due to various reasons, such as physical limitations and lack of motivation. Clinical studies suggested the efficacy of WBV to counteract loss of muscle strength and power in the elderly [23,40–44], where WBV induces greater neuromuscular activity [45], increases the cross-sectional area of Vastus Medialis (VM) and Biceps Femoris (BF) muscles [46], and results in increased countermovement jump height after training [31,47] in the elderly.

### 3.2. Clinical Studies on Effects of WBV on Muscle with Postures Other Than Standing

Most clinical studies reporting improved muscle power/strength were conducted with superoinferior (vertical) vibration applied to the feet of a standing body, while very few studies have been reported with postures other than standing, such as sitting, supine, or prone. Clinical studies investigating the effects of vibration on muscle with postures other than standing are summarized in Table S3. Although clinical studies have suggested the capability of WBV to mitigate the loss of muscle strength and power in the elderly, as alluded to in the previous sections, WBV in standing posture with flexed knees for a period of time in multiple sessions of treatment may be too challenging physically, particularly to the frail elderly who are in greatest need of improved muscle strength. Vibration in sitting or supine positions would be easier and may improve the adherence and feasibility of the intervention in the elderly population. Greater change in EMG (rms) value was reported with vibration in positions other than standing, for example, in Vastus Medialis (VM) and Vastus Lateralis (VL) muscles, respectively, after vibration treatment in semi-supine position [48], and Vastus Lateralis (VL) and Biceps Femoris (BF) muscles in sitting position [49]. The underlying mechanisms regarding how WBV increases neuromuscular performance as a form of EMG (rms) value are not clear [4,8]. Nevertheless, it is believed that TVR is responsible for increased muscle activity, i.e., the increase in EMG (rms) values, during WBV [4,7,8]. Vibration in prone and supine positions with slow-velocity resistance training was more effective in improving muscle power than with body-weight exercises [48,50]. As the outcome parameters used in the clinical trials for measuring muscle strength are different for different postures, it is difficult to compare the effects of vibration between standing and other postures.

### 3.3. Adverse Effects of Vibration on Human Body

Although a body of evidence indicates that vibration may improve muscle power and strength, exposure to mechanical vibration may create adverse effects on the human body [51,52]. In order to determine an optimal vibration protocol that is most beneficial to human health, adverse effects of vibration must be investigated and be avoided clinically. High-frequency vibration to the soft tissues can increase shear stress and may increase the risk of injury in peripheral, neural, vascular, and musculoskeletal systems. The adverse health effects associated with WBV or hand-arm vibration and frequency-dependent effects of vibration have been reviewed by Krajnak et al. [51] Increased stress and strain in tissues due to vibration (>100 Hz) can lead to vascular and sensorineural dysfunctions [51]. Vibrations in the range of 100–150 Hz result in greater TVR which produces higher muscle stress and fatigue [53]. Low-frequency vibration (20 Hz) may be responsible for abdominal and chest pain, discomfort feeling, and head and lower jaw symptoms [54]. WBV is also associated with development of metabolic disorder [55] and chronic diseases including cardiovascular diseases [56]. Table S4 presents a summary of literature reporting adverse effects of vibration on humans.

### 3.4. Optimal Vibration Protocol

Optimal protocols for vibration treatment that can elicit maximum health benefits are always desirable. Some studies [13,35,57–59] tried to address the optimal protocols for WBV

treatment. Although optimal protocols were suggested by these studies, no true optimal protocols were established scientifically. As the results of those studies were totally based on clinical trials, all WBV intervention parameters were fixed in each study, with frequency, amplitude, knee flexion angle, exposure time, and/or additional load during WBV all set at discrete values. Some other parameters such as age of the participants, muscle type, body posture, location of vibration source can also influence the optimal conditions. Therefore, a clinical study may report an optimal vibration protocol selected from a few treatment protocols applied to its subject groups only for that study, but it is possible that none of the treatment protocols implemented in that study was close to the true optimal protocol. In order to find the true optimal condition, all the aforementioned WBV parameters need to be taken into account as long as evidence shows that the parameters influence muscle behavior. Discussion on the optimal WBV protocol based on the present literature review is presented in the section below.

## 4. Discussion

The effects of WBV intervention on muscle health depend on multiple parameters, making it difficult to analyze through mathematical modeling and to determine the optimal vibration protocol. Effects of WBV on muscle health may depend on vibration parameters (frequency, acceleration, and amplitude), vibration mode (superoinferior, transverse, rotational, etc.), the muscle, exposure time, type of exercise before or during vibration (weight carrying, squat, etc.), knee flexion angle (if squat), body position (standing, sitting, supine, prone, etc.) To our knowledge, this is the first review that attempts to find the optimal vibration protocol by reviewing clinical trials for improving muscle power and strength along with consideration of adverse effects. The following discussion is based on the literature review conducted in this study.

### 4.1. Optimal Vibration Protocol

Although evidence indicates that WBV results in improved muscle activity, little information is available in terms of optimal vibration protocol for the maximum neuromuscular and muscular strength benefits and minimal adverse effects. As mentioned above, effects of WBV on muscle health may depend on a number of parameters, and they should be considered in addressing the issue of optimal vibration protocol, but this is difficult to achieve with the existing data in the literature.

4.1.1. Optimal Vibration Frequency

Table S4 suggests that except for a few cases [53,60], adverse effects of vibration on human body occur at either very low frequencies (<20 Hz) or very high frequencies (>100 Hz). As the resonance frequency of the whole body lies in the low frequency range (<20 Hz), adverse effects arise from the high transmissibility of vibration in this frequency range. On the other hand, a high frequency (>100 Hz) causes greater muscle stress which can lead to damage in the soft tissues. In addition, frequencies higher than 60-Hz may develop symptoms of Hand-Arm Vibration Syndrome (HAVS) [61,62]. In order to avoid these adverse effects, most of the clinical studies applied WBV in the frequency range of 20 Hz to 60 Hz. It was found that WBV at 30 Hz frequency and 4-mm peak-to-peak amplitude for 8 weeks can improve the voluntary isometric contraction and concentric peak torque of knee extensors of untrained women [14]. Vibration at 30 Hz frequency and 2–4 mm of amplitude provides greater effectiveness for muscle improvement than any higher frequency and greater amplitude [17–19,63,64]. On the other hand, 50 Hz frequency with peak-to-peak amplitude of 3 mm and 2.51 mm can enhance the performance of 1RM (repetition maximum) squatting performance [20,21]. In addition, Petit et al. [59] demonstrated that WBV training with 50 Hz frequency can improve the eccentric peak torque by approximately 16.3% above the baseline. Yang et al. [65] applied dual frequency WBV (35 Hz and 45 Hz) in different directions (vertical, horizontal) and observed better muscle activity and enhanced change-of-direction (COD) ability.

As shown in Tables S1 and S2, vibration at 30 Hz has been employed in more WBV studies than any other frequencies. However, selection of 30 Hz has not been scientifically justified by plausible rationale in these clinical studies, and 30 Hz has not been scientifically established as the true optimal frequency for WBV. Among the 17 studies included in Table S1 for clinical studies of WBV on muscle in young adults, 9 studies tested with 30 Hz WBV. Among these 9 studies, 5 studies [16,19,33,34,64] tested only WBV at 30 Hz and did not test any other frequencies. Although these studies all demonstrate beneficial effects provided by 30 Hz WBV, they provide no evidence that 30 Hz performs better than other WBV frequencies. Among the 4 studies that tested other frequencies in addition to 30 Hz, only one study [12] tested multiple frequencies (5–30 Hz) and reported that 30 Hz is the best frequency in terms of muscle power and strength. However, two studies [20,59] both tested 30 Hz and 50 Hz, and 50 Hz performs better than 30 Hz. In studies [12,65] 30 Hz was the highest frequency tested and was found to perform better than any lower frequencies tested in the same study, but 30 Hz should not be taken as the optimal frequency because larger beneficial effects may occur at frequencies higher than 30 Hz based on the data trend, although no higher frequencies were tested in those studies. Therefore, studies [12,20,59,64] all suggest that an optimal frequency should be at or greater than 30 Hz. On the other hand, study [57] is the only study reporting an optimal frequency at 45 Hz based on a test of 20–55 Hz.

In Table S2 summarizing WBV studies in the elderly, the data of [66] demonstrated a trend of muscle activity increasing monotonically with frequency based on tests at 6, 12, 18, 24, and 30 Hz. Among these frequencies, the test data showed the largest muscle activity at 30 Hz, but 30 Hz should not be taken as the optimal frequency because larger muscle activity may occur at frequencies higher than 30 Hz, although not tested in that study. Krol et al. [67] reported a maximum change of EMG (rms) up to 86.91% and 73.55% (Table S1, Figure 2) at 60 Hz for VL and VM muscles, respectively, higher than that of 20 Hz and 40 Hz. Table S1 and Figure 2 indicate that the maximum change in EMG value increases with frequency almost monotonically in all six muscles (VL, VM, BF, RF, SOL, and GM). In all the plots showing maximum change in EMG vs. frequency in Figure 2, data points connected with solid lines represent data collected under exactly the same experimental condition (same amplitude and knee flexion angle), except changes only in the frequency. In summary, published data suggest that the optimal WBV frequency may be greater than 30 Hz although 30 Hz has been employed in more clinical studies than any other frequency. The result supported by more clinical studies than anything else is that the beneficial effects of WBV increase with frequency [12,20,31,59,64,67–70], but the optimal frequency has not been established.

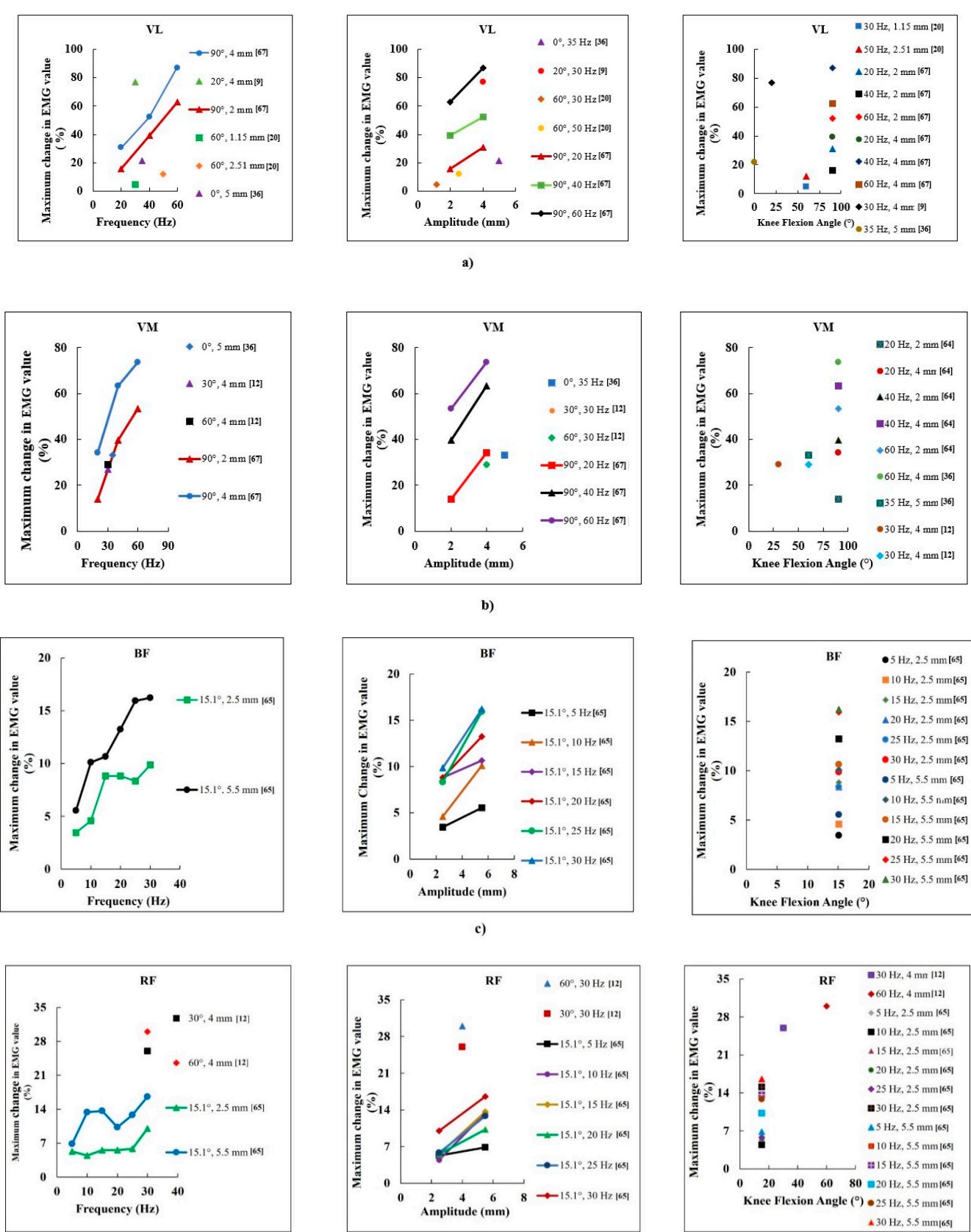

**Figure 2.** *Cont.*

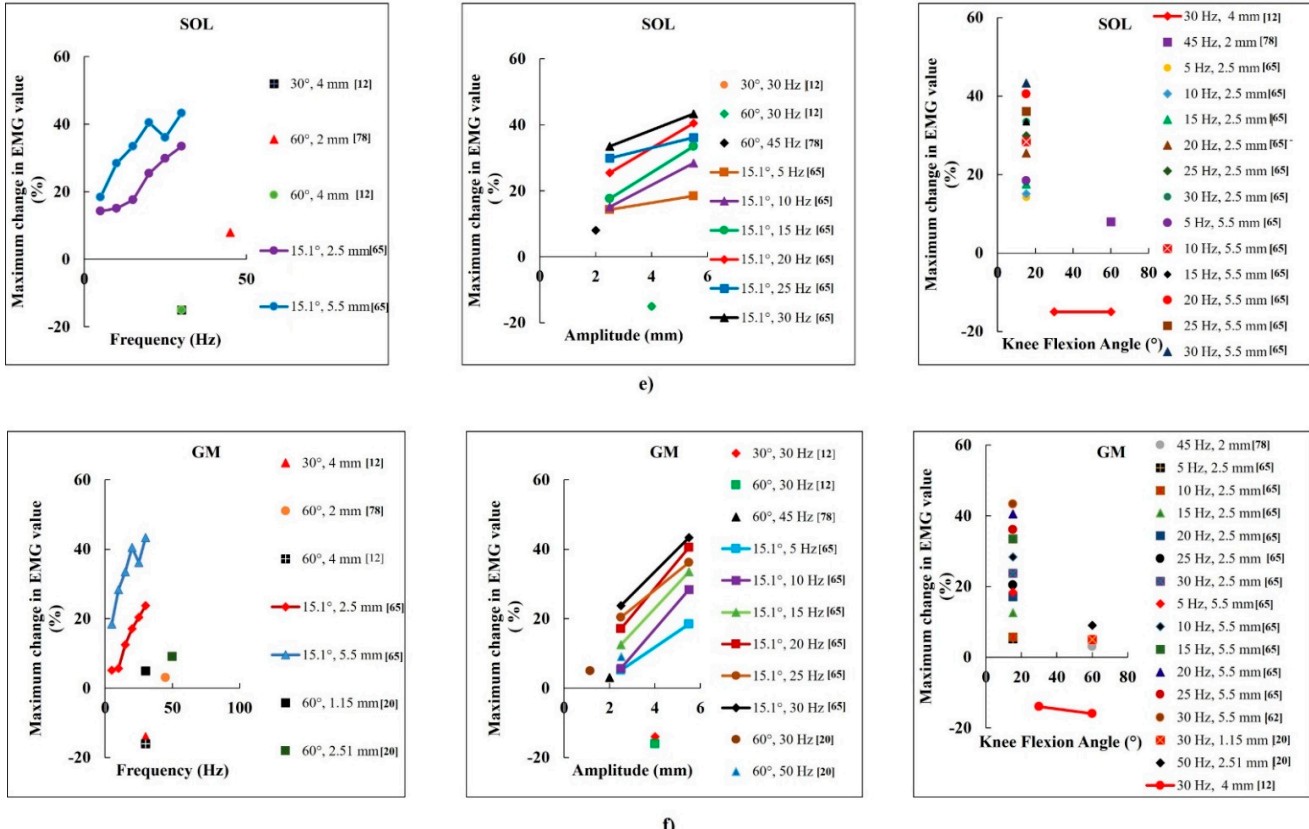

**Figure 2.** Maximum change in EMG (rms) value with frequency, knee flexion anlge, and amplitude for
(**a**) VL, (**b**) VM, (**c**) BF, (**d**) RF, (**e**) SOL, and (**f**) GM muscles, whole body vibration in standing position
with knee flexed and vibration source at the feet. Data points connected with solid lines represent
data taken under exactly the same experimental condition with only one parameter changed.

### 4.1.2. Optimal Amplitude

As shown in Tables S1 and S2, vibration at 4 mm amplitude has been employed in
more WBV studies than any other amplitude and has been demonstrated to bring forth
more beneficial effects on muscle than any other amplitudes tested in the published clinical
studies. However, the selection of 4 mm amplitude has not been scientifically justified by
plausible rationale in these clinical studies, and 4 mm amplitude has not been scientifically
established as the true optimal amplitude for WBV. Among the 28 studies included in
Tables S1 and S2 for clinical studies of WBV on muscle, 11 studies tested with 4 mm
amplitude. Among these 11 studies, 6 studies [9,12,19,40,47,58] tested only WBV at 4 mm
amplitude and did not test any other amplitudes. Although these studies all demonstrate
beneficial effects provided by WBV at 4 mm amplitude, they provide no evidence that 4 mm
amplitude performs better than other amplitudes. The other 3 studies [59,67] all tested
two amplitudes, 2 mm and 4 mm. All of those studies [59,67,70] show better performances
at 4 mm amplitude than 2 mm tested in the same study, but 4 mm amplitude should not be
taken as the optimal amplitude because larger beneficial effects may occur at amplitudes
even larger than 4 mm based on the data trend. However, larger amplitudes were not
tested in those studies. Pollock et al. [64] reported better performance at 5.5 mm amplitude
than 2.5 mm. Figure 2 (maximum change in EMG values vs. amplitude plot) also shows a
monotonic increasing trend in the maximum change of EMG (rms) value with amplitude
when frequency and knee flexion angle remain constant, up to 5.5 mm for BF, RF, SOL, and
GM and 4 mm for VL and VM, and an optimal amplitude is not indicated by the existing
data. In summary, published data suggest that the optimal WBV amplitude may be greater
than 4 mm although 4 mm has been employed in more clinical studies than any other

amplitude. Studies [20,59,64,67,70,71] support a trend that muscle activity increases with amplitude, but the optimal amplitude has not been established.

### 4.1.3. Effects of Knee Flexion Angle

Almost every clinical study of WBV adopts knee joint flexion during WBV for both young and old participants, as summarized in Tables S1 and S2, while studies involving senior adults adopt smaller knee flexion angles than those involving young adults, apparently due to physical limitations in the former. ISO health standards suggest that small knee flexion angles during WBV cause negative effects and should therefore be avoided [9,72]. Several studies tested with fixed knee flexion angles ranging from 15° to 100°, and reported significant increases in the activity of knee extensors [25,42,73], although no comparative analysis of muscle strength was conducted by any study. There is no evidence showing that neuromuscular response (in terms of sEMG-rms values) increases or decreases with knee flexion angle, because no WBV experiment has been performed with fixed test conditions (fixed vibration frequency, fixed amplitude, etc.) while changing only the knee flexion angle in order to investigate its effect, as shown in Figure 2 with a plot of maximum change in EMG value vs. knee flexion angle during WBV for each of 6 muscles. Carlucci et al. [57] reported that the sEMG-rms values of Vastus Lateralis (VL) muscle were higher at a knee joint angle of 90° compared to 120°, but without specifying the frequency tested for each angle. Ritzmann et al. [12] compared the results for 30° and 60° knee flexion angles and reported insignificant differences in EMG activity in Rectus Femoris (RF), Vastus Medialis (VM), Gastrocnemius Medialis (GM), and Soleus (SOL) muscles. It is thus concluded that the existing data do not suggest a clear effect of knee flexion angle during WBV on neuromuscular response; perhaps such effect is weak compared with the effects of WBV frequency and amplitude.

During WBV in standing position, significant vibration may reach the head and causing undesirable negative health effects. We should not just focus on the benefits of WBV on muscles without considering such adverse effects on the brain. The level of head acceleration varies with knee flexion angle. Abercromby et al. [9,16] established a relationship between knee flexion angle and the undesirable head acceleration due to vibration, and found that head acceleration reaches a maximum at 10° knee flexion angle, decreases toward a minimum at the knee flexion angle of 30°, and then increases with the angle [16]. This suggests that the ability of lower leg muscles to dampen mechanical vibration decreases for knee flexion angles greater than 30°. Increased head acceleration above the knee flexion angle of 30° can result from higher baseline neuromuscular activation influencing the joint compliance [16]. Although Abercromby et al. [16] mentioned that the lowest risk of adverse effects (in terms of head acceleration and mechanical impedance) is at around 26° to 30° knee flexion angle, knee flexion angles greater than 40° were not examined. In order to determine the optimal knee flexion angle for the maximum beneficial effect on muscles due to WBV, more data are needed about the knee flexion angle that can minimize head acceleration.

### 4.1.4. Effects of Posture and Muscle Type

Effects of WBV on muscle strength for positions other than standing is important considering the large elderly, frail, and disabled population in need of muscle strengthening therapy but cannot stand due to physical limitations. Among positions other than standing, studies with sitting positions, for instance, focus mostly on the adverse effects as summarized in Table S4. According to Pujari et al. [49], BF muscle benefits more from WBV than VL muscle at sitting posture (90° knee flexion angle) and 50 Hz frequency. The phenomenon can be associated with muscle tension under vibration. BF is a muscle in the hamstrings (posterior), which flexes the knee joint, and VL is a muscle in the quadriceps (anterior), which extends the knee. When the knee is flexed, BF contracts and generates tension, which results in higher muscle activity [67]. VL and VM, both muscles in quadriceps, have similar changes at 60 Hz and 90° knee flexion angle [67] as shown by red triangle (▲) and blue

circle (●) in maximum change in EMG value vs. frequency plot of both Figure 2a,b. Figure 2 also shows that at 30° knee flexion angle, 4-mm amplitude, and 60° knee flexion angle, 4 mm, VM and RF, both quadriceps muscles, benefit from WBV (see Figure 2b,d, frequency plots), while GM and SOL, both tibia muscles, do not (see Figure 2e,f, frequency plots) [12]. The data with sitting [49] (or any posture other than standing) seem to show a better muscle response than standing. However, there are no data reported directly comparing WBV in standing with any other posture such as sitting under exactly the same test condition (frequency, amplitude, knee angle, population, etc.). Some studies demonstrated that static squat training followed by dynamic squat training (squatting involving joint movement) with WBV can induce greater muscle force, and it is anticipated that the amount of muscle force due to dynamic squat with WBV is greater than that due to static squat with WBV [19].

### 4.1.5. Vibration Mode

Abercromby et al. [16] and Ritzmann et al. [12] examined the effect of vibration mode (rotational/side-alternating and vertical/synchronous) on the EMG activity while standing, and found that neuromuscular activity is greater during side-alternating vibration. In addition, side-alternating vibration allows more peripheral blood flow, muscle oxygenation, and increased heart rate compared to vertical (superoinferior) vibration [74]. Undesirable transmission of vibration to the head is 71% to 189% higher for vertical (superoinferior) vibration than side-alternating vibration, suggesting a higher risk of adverse health effects [9]. In addition, side-alternating vibration can create two times greater acceleration and less damping effects at ankle joints compared to vertical/synchronous vibration [75–77]. Mechanical induction in the muscle is more significant during side-alternating vibration than synchronous vibration by contributing to 66–91% higher EMG [12], as side-alternating vibration allows more mechanical stimulation as well as greater muscle activation intensity compared to superoinferior or vertical vibration [12]. Although most of the existing commercial WBV machines feature vertical (superoinferior) vibration and most of the reported studies investigate vertical (superoinferior) WBV, side-alternating WBV deserves more attention as it may provide more neuromuscular benefits.

### 4.2. Future Research

As muscle strength is affected by many parameters (physiological, neurological, biomechanical, etc.) that vary from person to person, optimal vibration conditions may need to be developed for each individual. A clinical study may demonstrate an optimal vibration protocol selected among a few treatment protocols applied to its subject groups, but it is possible that none of the treatment protocols implemented is close to the true optimal protocol. Future research should investigate the head acceleration or vibration transmissibility under different WBV conditions. One or two-dimensional lumped models have been established to investigate the dynamic interactions between the body and the vibration platform, while a three-dimensional model with continuous elements can provide more valuable information about the frequency response output parameters at each point of the body structure. However, lumped models do not consider the knee and hip flexion angles. Finite element human modeling can help understand the responses of different muscles at different knee and hip angles under simultaneous vibration and static or dynamic squatting. In particular, three-dimensional finite-element models including simulation of musculoskeletal tissues would better estimate the transmission of vibration to various muscles and anatomical elements of the body such as the head in real-time, including prediction of transmissibility under different vibration modes (superoinferior, anteroposterior, lateral, rotational, etc.) that may or may not create adverse health effects. The main disadvantages of a three-dimensional finite element model are long computation time, need for patient-specific geometry, and exact mechanical properties of all parts of the human body. In addition, three-dimensional models are unable to incorporate the physiological changes in response to muscle contraction. Perhaps data driven models can be employed to quantify the physiological changes related to WBV. As muscle force is

related to neuromuscular and physiological activities, a computational model needs to be developed to quantify the changes in neuromuscular and physiological functions of different muscles for a wide range of vibration parameters (frequency, amplitude, knee angle, posture, vibration mode, etc.). The model needs to predict the changes in motor unit threshold and firing rate, release of calcium from the sarcoplasmic reticulum, and complex overlapping of actin and myosin under vibration at different conditions. As the responses of the neuromuscular system change with external stimulation, it is important to investigate and understand the physiological as well as functional changes of the muscle fiber with vibration.

## 5. Conclusions

Based on a review of published studies, our analysis of existing data on the effects of WBV on different quadriceps and leg posterior muscles suggests that the beneficial effects of WBV increase with frequency, but an optimal frequency has not been established. Muscle activity increases with WBV amplitude, but an optimal amplitude has not been established. The existing data do not suggest a clear effect of knee flexion angle during WBV on neuromuscular response; perhaps such an effect is much weaker compared with the effects of WBV frequency and amplitude. Since existing data suggest that the beneficial effects of WBV increase with both frequency and amplitude, the optimal WBV protocol should be determined by taking into account the adverse effects of WBV on all parts of the human body including those related to head acceleration. WBV under sitting or postures other than standing deserves more research attention as it may provide a better muscle response than standing. More studies are also needed to investigate WBV of different modes such as anteroposterior as it may provide more neuromuscular benefits than superoinferior/vertical WBV.

**Supplementary Materials:** The following supporting information can be downloaded at: https://www.mdpi.com/article/10.3390/biomechanics2040043/s1, Table S1: Summary of clinical studies of effects of WBV on muscles in healthy young adults (18 to 32 years of age); Table S2: Summary of clinical studies of effects of WBV on muscle in elderly adults (54–82 years of age); Table S3: Summary of clinical studies on the effects of vibration on muscle with postures other than standing; Table S4: Summary of studies on the adverse effects of WBV. References [78–90] are cited in the supplementary materials.

**Author Contributions:** Conception and design: A.A.M., M.-C.C., C.-L.S. Data collection: A.A.M. Analysis and interpretation of results: A.A.M., M.-C.C., C.-L.S. Draft manuscript preparation: A.A.M., M.-C.C., C.-L.S. All authors have read and agreed to the published version of the manuscript.

**Funding:** The authors received no financial support for the research, authorship, and/or publication of this article.

**Institutional Review Board Statement:** Not applicable.

**Informed Consent Statement:** Not applicable.

**Data Availability Statement:** No new data was generated during the process.

**Conflicts of Interest:** All authors declare that they have no conflict of interest.

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
