# Peer review of "On the Optimal Whole-Body Vibration Protocol for Muscle Strength"

_2673-7078, doi:10.3390/biomechanics2040043_

Round 1
Reviewer 1 Report
Dear Authors,
This paper presents a review of studies on the effects of whole body vibration (WBV) on muscles and an analysis of data to identify the optimal protocols for the most beneficial neuromuscular responses. In general, the findings suggest that: I) the beneficial effects of WBV increase with frequency and amplitude; II) the optimal WBV protocol should be determined by considering the adverse effects of WBV on all parts of the human body. Directions for future research are discussed with regard to establishing the optimal WBV protocol as a safe and effective therapeutic/exercise modality for improving muscle strength and health. The topic of this study is very interesting and it provides important findings. However, the study presents specifics problems which must be corrected or clarified.
Abstract: The justification and purpose of the study are well informed. However, important methodological aspects need to be informed (for example, how many scientific journals was considered, inclusion and exclusion criteria, etc...). In addition, it is interesting to point out some quantitative results.
Keywords: List keywords in alphabetical order. We suggest adopting terms from the Health Sciences Descriptors (https://decs.bvsalud.org). Also, adopt keywords that are not written in the title.
Introduction: The introduction is relevant. However, a greater approach to the relationship between the justification and the objective of the study is suggested.
Page 1, line 47: “most beneficial muscular health outcomes” What do the authors consider to be the most beneficial results for muscle health? Neuromuscular prevention or performance?
Page 2, lines 52-54: “true optimization cannot be achieved by discrete and fixed values of parameters assigned to different groups tested in clinical trials” Why can't true optimization be achieved by discrete, fixed values of parameters assigned to different groups tested in clinical trials? Justify this statement and cite some bibliographic references.
Page 2, lines 55-59: “The present study aims to analyze the results of clinical trials in which WBV was applied to the feet of human subjects in both standing and sitting positions, as well as other postures (supine, prone), in an attempt to identify the optimal WBV intervention protocol that is safe while maximizing improved muscle strength” There is no clear relationship between the purpose and justification of the study. It is suggested to simplify the objective and detail the methodological aspects in the methodology section.
Methods: Systematic reviews, as the name implies, typically involve a detailed and comprehensive plan and search strategy derived a priori, with the goal of reducing bias by identifying, appraising, and synthesizing all relevant studies on a particular topic. The meta-analysis is a statistical analysis that combines the results of multiple scientific studies. Meta-analyses can be performed when there are multiple scientific studies addressing the same question, with each individual study reporting measurements that are expected to have some degree of error. Objectively, a systematic review answers a defined research question collecting and summarizing all empirical evidence that fits pre-specified eligibility criteria. The meta-analysis is the use of statistical methods to summarize the results of these studies. Is this a systematic review or meta-analysis?
Figure 1: Inform the period of publications. Figures and tables must be self-explanatory.
Results: The approaches adopted in the results are relevant. However, for each topic it is suggested to include a more quantitative breakdown and a more detailed conclusion of the main findings.
Page 3, lines 113-114: “2 to 4 sessions of training per week” What is the average time of sessions?
Page 4, lines 126-127: “Although resistance training can counteract age-related muscle strength loss” Can the resistance training counteract or retard the age-related muscle strength loss?
Page 4, lines: 133-134: “in the elderly” But why not in adults? What are the specifics in the neuromuscular processes of the elderly and adults? Neuromuscular economics, for example (Neuromuscular economy, strength, and endurance in healthy elderly men. J Strength Cond Res. 24(x): 000–000, 2010)?
Page 4, line 149: “Greater change in EMG” Are the changes a consequence of a neural adaptation or an increase in muscle mass? Furthermore, it is known that the individualized whole-body vibration can increased serum hormonal concentrations, muscle damage, and inflammation levels similar to those induced by resistance training and hypertrophy exercises.
Page 4, lines 157-158: “it is difficult to compare the effects of vibration between standing and other postures” In this case, an inverse dynamics approach can help (see, for example, Influence of whole body vibration on drop jump landings and knee loading mechanics - Journal of Orthopedics and Orthopedic Surgery, 2022).
Page 4, lines 160-162: “Although a body of evidence indicates that vibration may improve muscle power and strength, exposure to mechanical vibration may create adverse effects on the human body” Cite, please, one or more bibliographic references.
Discussion: Evidence that the vibratory stimulus is capable of increasing neuromuscular activity has been demonstrated by studies involving surface electromyography. Although there is no consensus about the mechanisms by which the vibratory stimulus affects the neuromuscular system, it has been suggested that the cause of the increase in motor unit recruitment is an excitatory response of the muscle spindles, due to the stretch reflex mechanism. Too, it has been demonstrated that neurophysiological factors involved in the response to vibratory stimulus have an important contribution of the oscillation frequency at which body structures are exposed. Despite its wide use in sports training and patient rehabilitation, there is still controversy regarding the factors that initiate neurophysiological responses in skeletal muscle during WBV. Furthermore, in everyday practice, therapists and practitioners promote longer WBV exposures, although the effects of such exercise modalities are mostly unknown. By increasing the WBV stimuli duration, it is suggested that WBV may acutely induce fatigue rather than potentiation.
This discussion is relevant. However, very descriptive and little speculative in relation to neuromuscular issues. A greater neuromuscular explanatory approach is suggested (see, for example: Effects of Whole Body Vibration on the Neuromuscular Amplitude of Vastus Lateralis Muscle - J Sports Sci Med. 2017 Sep; 16(3): 414–420; Effects of whole- body vibrations on neuromuscular fatigue: a study with sets of different durations - PeerJ. 2020; 8: e10388; Effects of strength, endurance, and concurrent training on aerobic power and dynamic neuromuscular economy in elderly men. J Strength Cond Res 25(3): 758–766, 2011).
Page 5, lines 211-214: “As mentioned above, effects of WBV on muscle health may depend on a number of parameters, and they should be considered in addressing the issue of optimal vibration protocol, but this is difficult to achieve with the existing data in the literature” However, based on the studies, what are the authors' hypothesis? It is important that the authors indicate some hypothesis, otherwise this topic does not become relevant.
Page 5, lines 218-2022: “As the resonance frequency of the whole body lies in the low frequency range (< 20 Hz), adverse effects arise from the high transmissibility of vibration in this frequency range. On the other hand, a high frequency (> 100 Hz) causes greater muscle stress which can lead to damage in the soft tissues” What could be the neuromuscular justifications?
Page 6, lines: 276-279: “The result supported by more clinical studies than anything else is that the beneficial effects of WBV increase with frequency, but the optimal frequency has not been established” What could be the neuromuscular justifications?
Figure 2: Please inform the origin of the data (figures must be self-explanatory).
Page 8, lines 314-316: “In summary, published data suggest that the optimal WBV amplitude may be greater than 4 mm although 4 mm has been employed in more clinical studies than any other amplitude” What could be the neuromuscular justifications?
Page 9, line 319 and 362: “Effects of knee flexion angle” and “Effects of posture and muscle type” What is the purpose of investigating the influence of WBV on biomechanical aspects? Postural correction, prevention of injuries, improvement in performance?
References:
2020-2022: 1% reference
2019-2018: 13% references
2017-2016: 19% references
2015-2012: 22% references
> 2011: 45% references
Note: PubMed: 10 results 2020-2022 (neuromuscular and vibration body). It is suggested to adopt more current references...
Author Response
Thanks for your review. Your points were really pertinent to the topic and useful to make the draft better. We tried to respond to your questions.
Previously, the format of the journal was making Figure 2 unreadable due to a very squeezed layout. Now for Figure 2, we have provided a figure with a better resolution.

Reviewer 2 Report
Main comments
The paper presents a review of publications concerned with the effects of whole-body vibration on muscle strength. The aim was to establish a protocol for determining an association between whole-body vibration and muscle strength. I offer my comments on the manuscript.
The line numbers below correspond to those on the draft manuscript.
Comments
Line 71
Change “… extrusion …” to “… exclusion …”.
Line 91
Change “… less …” to “… fewer …”.
Line 226
As the frequency is known (30 Hz) and the displacement is known (4 mm peak-to-peak), the acceleration magnitude (71 ms-2 r.m.s.) should be included. This should be included for other frequencies and displacement amplitudes (lines 228 and 231).
Line 259
Change “… data 20 …” to “… data covering the frequency range 20 …”.
Lines 280-283
The figure is not legible.
Lines 324-325
It is stated that “ISO health standards suggest that small knee flexion angles during WBV cause negative effects and should therefore be avoided [69]”. Reference is made to ISO 2631-1:1997. Please identify the section of the standard that states this; I could not find any confirmation for this statement.
Lines 416-418
Substantial research exists on the transmission of vibration to the head for seated and standing people. I suggest you look at chapter 8 in ‘Handbook of Human Vibration’ by MJ Griffin (1990) for more information.
Line 436
Change “… model …” to “… models …”.
Lines 482-483
Ensure format of text is the same as other references.
Line 602
Is this a made-up reference: H. Body, V. Exposure, and I. T. S. Measurement, “Human Body Vibration Exposure and Its Measurement,” 2009?
Line 642
The reference should be complete and show ISO 2631-1:1997.
Lines 663-664
Ensure format of text is the same as other references.
Line 690
References in Table S4 are not included in the references section: references 79-86.
Author Response

(The authors gave the same response as above.)

Round 2
Reviewer 1 Report
Dear Authors,
This paper presents a review of studies on the effects of whole body vibration (WBV) on muscles and an analysis of data to identify the optimal protocols for the most beneficial neuromuscular responses. In general, the findings suggest that: I) the beneficial effects of WBV increase with frequency and amplitude; II) the optimal WBV protocol should be determined by considering the adverse effects of WBV on all parts of the human body. Directions for future research are discussed with regard to establishing the optimal WBV protocol as a safe and effective therapeutic/exercise modality for improving muscle strength and health. The topic of this study is very interesting and it provides important findings.
1. Methods: Systematic reviews, as the name implies, typically involve a detailed and comprehensive plan and search strategy derived a priori, with the goal of reducing bias by identifying, appraising, and synthesizing all relevant studies on a particular topic. The meta-analysis is a statistical analysis that combines the results of multiple scientific studies. Meta-analyses can be performed when there are multiple scientific studies addressing the same question, with each individual study reporting measurements that are expected to have some degree of error. Objectively, a systematic review answers a defined research question collecting and summarizing all empirical evidence that fits pre-specified eligibility criteria. The meta-analysis is the use of statistical methods to summarize the results of these studies. Is this a systematic review or meta-analysis?
Response: Thanks for your questions. We classified our draft as narrative review. We reviewed the existing literatures on the effect of WBV on neuromuscular health and tried to identify the knowledge gap of this intervention.
Reviewer: You classified our draft as narrative review. So, I suggest including this information.
2. Results: The approaches adopted in the results are relevant. However, for each topic it is suggested to a more detailed conclusion of the main findings.
Response: Thanks for your suggestion. We have included as many quantitative breakdowns and conclusions as scientifically possible.
Reviewer: For each topic it is suggested to a general conclusion of the main findings.
3. Page 4, line 149: “Greater change in EMG” Are the changes a consequence of a neural adaptation or an increase in muscle mass? Furthermore, it is known that the individualized whole-body vibration can increased serum hormonal concentrations, muscle damage, and inflammation levels similar to those induced by resistance training and hypertrophy exercises.
Response: “The underlying mechanisms by which WBV affects EMG and enhances neuromuscular performance have yet to be established. However, it is strongly believed that increases in muscle activity, i.e., the presence of EMG, during WBV can be attributed to tonic vibration reflexes or stretch reflexes. Improvements in muscle strength and power post-vibration can in turn be attributed to the increased muscle activity and increased gravitational loading of the musculoskeletal system as a result of vibration, leading to subsequent neurogenic/morphological adaptation and/or changes in concentrations of hormones, such as testosterone, cortisol, and growth hormone.”
Reference:
Alizadeh-Meghrazi, M., Masani, K., Zariffa, J., Sayenko, D. G., Popovic, M. R., & Craven, B. C. (2014). Effect of whole-body vibration on lower-limb EMG activity in subjects with and without spinal cord injury. The Journal of Spinal Cord Medicine, 37(5), 525-536.
M. C. A Al Masud, HY Luk, CL Shen, “Impact of Local Vibration Training on Neuromuscular Activity, Muscle Cell, and Muscle Strength: a Review,” Crit Rev Biomed Eng, vol. Doi:10.161, 2022.
Reviewer: Ok! I suggest including this information (briefly) in this topic, with citations.
Author Response
Thanks for your additional comments. Our responses can be found in the attached file (marked in red font). Please let us know if you have any other comments.
